# The Impact of a Family-Based Assessment and Intervention Healthy Lifestyle Programme on Health Knowledge and Beliefs of Children with Obesity and Their Families

**DOI:** 10.3390/nu14204363

**Published:** 2022-10-18

**Authors:** Dayna T. Te’o, Cervantée E. K. Wild, Esther J. Willing, Lisa E. Wynter, Niamh A. O’Sullivan, Paul L. Hofman, Sarah E. Maessen, José G. B. Derraik, Yvonne C. Anderson

**Affiliations:** 1Department of Paediatrics, Taranaki Base Hospital, Taranaki District Health Board, David Street, New Plymouth 4310, New Zealand; 2Department of Paediatrics: Child and Youth Health, Faculty of Medical and Health Sciences, University of Auckland, Private Bag 92019, Auckland 1142, New Zealand; 3Liggins Institute, University of Auckland, Private Bag 92019, Auckland 1142, New Zealand; 4Kōhatu—Centre for Hauora Māori, Division of Health Sciences, University of Otago, Dunedin 9016, New Zealand; 5Enable Institute, Faculty of Health Sciences, Curtin University, Kent Street, Bentley, WA 6102, Australia; 6Telethon Kids Institute, Northern Entrance, Perth Children’s Hospital, 15 Hospital Avenue, Nedlands, WA 6009, Australia; 7Community Health, Child and Adolescent Health Service, 2 Mill Street, Perth, WA 6000, Australia

**Keywords:** pediatric, obesity, community, randomized clinical trial, health promotion, health literacy

## Abstract

Objective: To determine the impact of a family-based assessment-and-intervention healthy lifestyle programme on health knowledge and beliefs of children and families affected by obesity. Second, to compare the health knowledge of the programme cohort to those of a national cohort in Aotearoa/New Zealand (NZ). Design: This mixed-methods study collected health knowledge and health belief data in a questionnaire at baseline and 12-, 24-, and 60-month follow-up assessments. Health knowledge over time was compared with baseline knowledge and with data from a nationally representative survey. A data-driven subsumption approach was used to analyse open-text responses to health belief questions across the study period. Setting: Taranaki region, a mixed urban–rural setting in NZ. Participants: Participants (caregiver/child dyads) from the Whānau Pakari randomised trial. Results: A greater proportion of the cohort correctly categorised foods and drinks as healthy or unhealthy at 12 months compared to baseline for most questionnaire items. Retention of this health knowledge was evident at 24- and 60-month follow-ups. More than twice as many participants correctly reported physical activity recommendations at follow-up compared to baseline (*p* < 0.001). Health knowledge of participants was similar to the national survey cohort at baseline, but surpassed it at 12 and 24 months. Participant beliefs around healthy lifestyles related to physical functioning, mental and emotional wellbeing, and enhancement of appearance, and gained greater depth and detail over time. Conclusions: This study demonstrates the important role that community-level healthy lifestyle programmes can have in knowledge-sharing and health promotion.

## 1. Introduction

Childhood obesity remains a major global health challenge that continues to place a significant health and financial burden on populations worldwide [1,2]. Because obesity is preventable, a multifactorial approach is needed with global and local strategies to alleviate the associated health burden on societies and financial burden on the health system [3].

The translation of health knowledge into positive health behaviours acts as a catalyst for improved health outcomes [4]. An essential component within this process is health literacy, a concept that first emerged in the 1960s to describe an individual’s ability to find, understand, and use information to make choices that affect their health. Previous research demonstrated an association between low health literacy of an individual and poorer health outcomes, including poor overall health status and higher mortality [5,6]. Health promotion combines health literacy with aspects such as the mobilisation of personal and societal influences and advocacy in policy making. These concepts work together to bring about effective health promotion [6]. The delivery of these three aspects of health promotion lessen the burden of childhood obesity on our communities and the health care system.

Previous studies showed that community-based health interventions can improve participants’ health outcomes, whether through direct measurement of health knowledge via surveys [7,8,9] or assessment of health behaviours [10,11,12]. However, there is scant research analysing health knowledge in healthy lifestyle programmes, where the mainstream intervention of a hospital-based paediatric appointment has been modified to bring care into the community through home-based assessments.

Whānau Pakari is a family-based assessment and intervention healthy lifestyle programme that supports children and adolescents (aged 4–16 years) experiencing weight issues in the Taranaki region of Aotearoa/New Zealand (henceforth referred to as New Zealand). By using methods such as home-based assessments and family-centred and community-based interventions, this programme aims to improve access to and acceptability of care for those most affected by obesity and its associated comorbidities (namely Māori and people from the most deprived households). Whānau Pakari has previously been evaluated by a randomised clinical trial [13,14,15] that provided the baseline and follow-up data on healthy eating, physical activity and health knowledge used in this study.

The aims of this study were: (a) to determine the impact of the Whānau Pakari programme on health knowledge and beliefs over time of children affected by obesity; and (b) to compare the health knowledge of the Whānau Pakari cohort at baseline to those of a population-based cohort within the New Zealand 2008/9 Nutrition survey.

## 2. Materials and Methods

Whānau Pakari was an unblinded randomised clinical trial that assessed the effects of a 12-month healthy lifestyle assessment-and-intervention programme in Taranaki, New Zealand in 2012–2015 [13,14,15]. Taranaki is a mixed urban–rural region, where Māori (New Zealand’s Indigenous people) comprise 17% of the population [16]. Participants were randomised into a high-intensity intervention or a low-intensity intervention group. The high-intensity group received holistic six-monthly home-based assessments and weekly group physical activity or nutrition sessions, while the low-intensity group received the home-based assessment-and-advice only. Advice came in the form of 5 key messages relating to caregivers being a role model in terms of eating and physical activity, eating more fruit and vegetables, checking portion size, being active every day in every way possible, and drinking water or milk [17]. The assessment was a weight-related assessment that included screening for associated comorbidities, dietary habits, and physical activity, and psychological screening [14]. Socioeconomic status was estimated based on the NZ Index of Deprivation 2006 (NZDep2006) [18]. In this study, NZDep2006 is reported in quintiles, with 5 indicating the participant resides in an area with high relative deprivation based on census data, and 1 indicating low deprivation [18]. Comparisons between the high- and low-intensity trial groups are published elsewhere [13,14,15] and are not a focus of the current study. For main analyses, all participants are considered to be one group, the ‘Whānau Pakari cohort’, which is deemed more representative of real-wold clinical service provision.

Eligibility criteria for the trial were children and adolescents between the ages of 4–16 years who had been identified as being affected by obesity (BMI ≥ 98th centile for age and sex), or overweight (BMI > 91st centile) with weight-related comorbidities, using a modification of the UK Cole data [19]. These cut-off values are accepted for use by the New Zealand Ministry of Health [20]. Children and adolescents were referred for participation by various community services such as public health nurses, paediatricians, general practitioners, social workers, and self-referrals. Participants who had co-existing medical or psychological conditions that would inhibit their ability to participate in the weekly physical activity and information sessions were excluded. Participants were also excluded if they did not have a caregiver committed to the programme or if participants were not considered ready to make healthy lifestyle changes. All Whānau Pakari participants who completed a health knowledge questionnaire were eligible for the current analyses, including some participants who were assessed but not randomised into the Whānau Pakari trial (reported elsewhere [13,14]).

Ethics approval was granted by the Central Health and Disability Ethics Committee (Ministry of Health, New Zealand; CEN/11/09/054), and the trial was registered with the Australian New Zealand Clinical Trials Registry (ANZCTR: 12611000862943). Written informed consent was obtained from all participants or their caregivers, or, where appropriate, verbal assent for younger children.

### 2.1. Whānau Pakari Quantitative Health Knowledge Data

For this study, data were collected at baseline, and at 12-, 24-, and 60-month follow-ups. A questionnaire based on the National Survey of Children and Young People’s Physical Activity and Dietary Behaviours in New Zealand 2008/09 (henceforth referred to as National Survey) [21] was completed by the participant and their caregiver face-to-face as a caregiver-child dyad at each time point. Because the questionnaire data was collected as part of a multi-disciplinary health service, this response style is in line with how questionnaires are completed in clinical practice, as well as with the methodology of the National Survey [21] (Appendix A). Respondents were asked to categorise certain foods and beverages as either “healthy”, “unhealthy”, “neither”, or “don’t know”. Plain milk, plain water, and all fruit and vegetables (frozen, fresh, and canned) were classified as ‘healthy’; flavoured milk, fizzy drinks, takeaways and fast foods, crisps and chips, sweets and lollies, and snack bars as ‘unhealthy’. This categorisation was used in the National Survey and is based on the NZ Ministry of Health’s Food and Beverage Classification System, which is intended as a practical guide to implementing food and nutrition guidelines in schools [21,22] Also included in the Whānau Pakari questionnaire was the question “What is the amount of physical activity you need to do to stay healthy, in minutes per day?”. Any answer equal to or greater than 60 min was categorised as a ‘correct’ answer, as per the NZ Ministry of Health guidelines for 5–17-year-olds [23].

### 2.2. Comparison of Whānau Pakari Data to National Data

Comparison was made between the Whānau Pakari cohort and respondents to the National Survey [21]. That survey of 5–24-year-olds was designed to be representative of the NZ population across regions, urban/rural status, age groups, and ethnicities based on census data. The themes available for comparison in the National Survey were as follows: 755 children aged 5–9 years and their caregiver who were asked to categorise food and beverages; and 1197 participants with children aged 10 to 17 years who were asked about recommended daily minutes of physical activity.

### 2.3. Whānau Pakari Qualitative Health Knowledge Data

Data were obtained from the two open-text questions relating to health knowledge in the same Whānau Pakari questionnaire. The two questions were “What are the benefits of healthy eating?” and “What are the benefits of physical activity?”.

### 2.4. Data Analyses

#### 2.4.1. Quantitative Analyses

Initially, health knowledge of Whānau Pakari participants at baseline was compared to their respective responses at the post-intervention follow-up assessments at 12, 24, and 60 months within each trial randomisation group. Differences in these changes were compared between trial randomisation group using two-sample McNemar’s tests [24]. After these initial analyses, all participants were treated as one cohort, which was more representative of the interactions of participants with a clinical service as a whole. Health knowledge of the study cohort at 12, 24, and 60 months was similarly compared to baseline using McNemar’s tests.

For comparisons with data from the National Survey [21], Whānau Pakari participants were stratified into two groups based on the age of the child participant (5–9 years and ≥10 years), with comparisons made to their health knowledge on food and beverages collected at baseline, 12 months, and 24 months. No data at 60 months were compared to national data, as our respective sample size for the food and beverages questionnaire was too small for reliable comparisons (*n* < 10). The outcomes assessed were the proportions of respondents in the two cohorts who correctly: (a) classified the specified food and beverages as either “healthy” or “unhealthy” (child participants aged 5–9 years); and (b) reported the minimum recommended minutes of physical activity per day (child participants aged ≥ 10 years). Potential differences were assessed with Chi-square tests or Fisher’s exact tests.

Data on all trial participants were included together in the analyses, as there were no observed differences in treatment outcomes between the high- and low- intensity groups [13]. Analyses were carried out using SPSS v27 (IBM Corp., Armonk, NY, USA), GraphPad Prism v8.4 (GraphPad Software Inc., San Diego, CA, USA), and Minitab v19 (Pennsylvania State University, State College, PA, USA). There was no imputation of missing data. Statistical tests were two-tailed, with the significance threshold at the 5% level.

#### 2.4.2. Qualitative Analyses

Given the brevity and specificity of the qualitative data, open text questions were analysed using an inductive qualitative content analysis approach. Data were coded by D.T.T. under supervision of C.E.K.W. and Y.C.A. The coding focused on the explicit content of the response. D.T.T. developed the initial coding framework, which involved both data summarisation and subsumption [25]. Coding of the data was completed using nVIVO Pro 12.x64 (QSR International, Melbourne, Australia). Data on the benefits of healthy eating were coded separately from physical activity data. No comparison to the national survey was possible due to lack of comparable qualitative data.

## 3. Results

239 child-caregiver dyads from the Whānau Pakari trial completed the health knowledge questionnaire at baseline and were included in the qualitative analyses. 161 dyads also had data for at least one follow-up time point for food knowledge and 157 for physical activity knowledge, and therefore were included in analyses comparing health knowledge over time. Table 1 outlines the demographic characteristics of child participants included in these analyses. There was strong Māori representation at the 12-, 24-, and 60-month follow-ups [*n* = 60 (41%), *n* = 51 (43%), and *n* = 37 (46%), respectively] for food knowledge items). There was also high representation from those living in the most deprived quintile of households (Table 1) in comparison to background rate in Taranaki (15% of the population) [26]. At all time points, the vast majority of accompanying caregivers were mothers (83–85%), with fathers making up an additional 10–12% (Table 1).

Demographic data for the two randomisation groups at each assessment are provided in Appendix A. The characteristics of the children and adolescents with physical activity data over time are provided in Appendix A, and data on participants compared to national survey data in Appendix A.

### 3.1. Quantitative Data: Change in Health Knowledge over Time

Improvements in health knowledge over time within randomisation groups are presented in Appendix A The only intervention effect was seen for physical activity duration, with more participants displaying improvements in knowledge in the high-intensity intervention compared to the low-intensity group at 12 months [38 (57%) vs. 18 (30%); *p* = 0.044] and 60 months [23 (58%) vs. 11 (34%); *p* = 0.045].

The Whānau Pakari cohort demonstrated improved health knowledge from baseline at each time point (Table 2). A larger proportion of participants correctly categorised food and beverage items following the 12-month intervention compared to baseline, and this improvement was sustained at the 24- and 60-month follow-ups for most items. For example, 88% of dyads (127 of 145) recognised ‘plain milk’ as healthy at 12 months compared to 72% at baseline (105 of 145; *p* = 0.001), while ≥93% of dyads recognised ‘crisps and chips’ as unhealthy at each follow-up [136 of 145, 112 of 120, and 78 of 81] compared to 83% at baseline (120 of 145; *p* = 0.001) (Table 2). Exceptions included ‘flavoured milk’ and ‘snack bars’, with an increased proportion of correct responses among participants who remained in the study at the 24- and 60-month follow-ups, but not at 12 months. These three items were also the least likely to be categorised accurately at baseline. An improvement in the knowledge of optimal daily physical activity was observed, with more than twice as many dyads able to correctly report the recommended number of minutes at each time point compared to baseline.

### 3.2. Health Knowledge in Whānau Pakari vs. National Survey

Table 3 outlines comparisons between the National Survey cohort and the Whānau Pakari cohort, whose demographic characteristics are provided in Appendix A. With the exception of ‘flavoured milk’, more than 80% of national survey respondents were able to correctly categorise each food and beverage item. At baseline the Whānau Pakari cohort and national survey respondents were similar in terms of healthy food and drink knowledge. However, the Whānau Pakari cohort was more likely than national survey respondents to correctly categorise ‘all fruit/vegetables’, ‘flavoured milk’, ‘takeaways/fast foods’ and ‘crisps/chips’ as healthy or unhealthy at 12 months, and ‘flavoured milk’ and ‘crisps/chips’ at 24 months.

At baseline, a smaller proportion of the Whānau Pakari cohort were able to correctly quantify the recommended daily minutes of physical activity compared to the national cohort. At the 12- and 24-month follow-ups, a greater proportion of Whānau Pakari participants demonstrated this knowledge than national survey respondents.

### 3.3. Qualitative Data: Benefits of Healthy Eating and Physical Activity

Because there was no direct comparison between participant responses at different time points, questionnaires from all 239 participants were coded for analysis at baseline. Responses at 12, 24, and 60 months are from the same participants described in Table 1. Table 4 outlines themes and subthemes identified from open-text questions relating to the benefits of healthy eating and physical activity in the health beliefs questionnaire.

Benefits of healthy eating and physical activity reported by participants were improved physical functioning, improved mental and emotional wellbeing, and enhanced outward appearance. Over time, there was an increase in the depth of answers, as well as a reduction in proportion of the cohort who answered with ‘don’t know’, indicating improvements and greater breadth of health knowledge relating to healthy eating and physical activity. At baseline, 30% of child-caregiver dyads (72 of 239) could not name any benefits of physical activity, and 20% (50 of 239) could not name any benefits of healthy eating. By 12 months, only 8% (12 of 145) and 14% (21 of 145) could not name any benefits of physical exercise or healthy eating, respectively, and at 24 months only 2% (3 of 122) could not name benefits of physical activity. The proportion of the cohort answering ‘don’t know’ to the healthy eating question at 24 months did not change further (14%; 17 of 122).

By the 60 month follow up all participants provided knowledge of a benefit of healthy eating and physical activity. This implies either newfound health beliefs or reinforcement of beliefs that had not been explored or expressed previously.

Enhanced physical functioning was a key benefit identified for both healthy eating and physical activity, specifically in relation to improvements in one’s energy levels and fitness capacity. Mental and emotional wellbeing benefits were acknowledged, including improved mood and reduced stress levels, as well as healthy self-perception and how participants believed the world perceived them as individuals. Improvements in one’s outer appearance was another common positive recognised by the cohort, including answers around weight loss and physical appearance. Socialisation with friends and peers was recognised as a benefit during the study. In comparison to the views expressed at baseline, by 60 months, greater recognition of health in relation to one’s own body and inner functioning was demonstrated, along with mental and emotional wellbeing-related benefits and less stigmatising answers when referring to one’s outer appearance.

## 4. Discussion

This study showed that the Whānau Pakari programme had a long-term positive impact (up to 5 years) on the health knowledge of children and adolescents affected by obesity and their caregivers. Compared to baseline, participants largely retained improvements in their recognition of dietary components that are healthy and unhealthy. Throughout the study, there was also a greater than 2-fold increase in the proportion of dyads recognising that ≥60 min of physical activity per day was recommended as per the guidelines at the time. Further, when compared with the most recently published National Survey of Children and Young People’s Physical Activity and Dietary Behaviours in New Zealand: 2008/09 [21], the Whānau Pakari cohort was able to demonstrate a greater level of health knowledge at 12 months and beyond. Notably, the greatest improvements in health knowledge were observed at 12 months during the time of the Whānau Pakari trial among participants in both low- and high-intensity groups, highlighting the benefits of involvement in healthy lifestyle programmes.

Interestingly, at baseline, 10% of Whānau Pakari participants and 13% of those aged 5–9 years categorised water as either ‘unhealthy’, ‘neither unhealthy or healthy’ or ‘don’t know’. Whilst these figures compare unfavourably to National Survey data and may initially appear surprising, anecdotal accounts indicate that a local township had been experiencing a compromised water source during the time of recruitment into this programme which, from the perspective of individuals in that community, may lead to the perception that water was unsafe and unhealthy to consume [27]. This highlights the importance of contextualising such findings with the environment families reside in. Setting is acknowledged as a key determinant of health which can create or solve problems for those within that setting [28]. Determining place of residence in relation to health knowledge was not within the scope of this study; however, it is a potential area for future research. Despite this initial finding at baseline, a positive change was seen at 12 months and 60 months, with 100% of the cohort identifying water as healthy.

As children navigate barriers to achieving healthy lifestyle change, development of positive health beliefs play an important role in motivating a child to make healthy lifestyle changes [29]. The health beliefs identified in the qualitative section of this study were varied and included benefits likely crucial to children and adolescents’ social and emotional wellbeing. Participants increasingly identified themes relating to positive experiences in self-esteem, mood, and social situations (e.g., participating in sports at school) as they progressed through the Whānau Pakari programme. By expanding the perceived health benefits beyond weight loss alone, motivation to maintain healthy lifestyle changes may be cultivated.

Other community-based health interventions have shown to have an overall positive impact on the health of participants [7,8,9,10,11,12]. The inclusion of Māori and Pacific facilitators was an important aspect of the Good Start programme in Queensland, allowing incorporation of culturally sensitive activities to best cater for those within the population who are most impacted by obesity [9]. Whānau Pakari was designed to provide an intervention that was effective for Māori families and those from most deprived households, who experience over-representation in obesity statistics in New Zealand [13,14]. These population groups are well represented within the Whānau Pakari cohort and face unique environmental and socioeconomic barriers to living in good health [30,31]. Translation of health knowledge into health behaviour change requires the elimination of societal and environmental barriers where possible, and adoption of facilitators to engagement, such as the provision of respectful, compassionate care [5,31]. Previous research acknowledges the key role that cultural appropriateness and sensitivity play in the design and delivery of an effective health intervention. Both the SHARE-AP ACTION study in Canada and the Good Start Programme in Australia purposefully tailor their intervention methodology in a way that best engages with their Indigenous population groups [5,9]. In this study, Māori participants comprised more than 40% of the cohort at each time point, demonstrating this programme’s ability to reach and engage Māori whānau.

Previous research has undertaken follow-up periods of up to one year, with varying levels of improved and sustained health knowledge with time. A strength of this study was the 60 months follow-up period. By measuring health knowledge 60 months from baseline, a more robust indication of whether this knowledge is retained can be assessed. A limitation of this study is the inability to determine the causality of improvements in health knowledge, which could be attributed to external sources of health information and the participant’s increased comprehension with age. It is also possible that whānau completing follow-up assessments were more engaged with the programme and motivated to increase their health knowledge than those who were lost to follow-up. The data showed that the greatest changes were seen at 12 months, following the period of time in which the participants had the greatest level of involvement with the programme and contact with the Whānau Pakari team.

Another aspect of the study that warrants caution in interpretation of results and comparison to other literature is the child-caregiver dyad approach to questionnaire completion. It has been used in health research to ensure collection of reliable data when the alternative is either proxy report or missing data [32]. A review of 7 studies using this dyad approach described its strengths in allowing for consistent methodology across a wide age range of child and adolescent participants [32], and it has been demonstrated to be more consistent with individual child report than an adult proxy in describing health-related experiences [33]. This approach is reflective of the Whānau Pakari programme as a real-world, family-focused programme. However, changes in child knowledge could be masked by parents answering the questionnaire who may not be as engaged as the child within the programme. This was also a limitation for the National Survey, where there was strong involvement of caregivers and parents in answering questions for younger children. In addition, the National Survey data were the most recently published data by the New Zealand Ministry of Health that analyses the health beliefs and knowledge of a cohort aged 5–19 years. It would be beneficial to repeat this national survey to understand current behaviours relating to nutrition and physical activity knowledge, and changes in knowledge over time. The nature of this study as an analysis of a real-world health service also limited the amount of data available for caregivers, for whom providing information about themselves during the assessment was not a requirement for participation. As a result, potentially important data such as caregiver’s educational attainment were not collected.

## 5. Conclusions

In conclusion, this study demonstrates improvements in health knowledge and beliefs among families participating in the Whānau Pakari programme that were sustained over time (up to 5 years). By modifying an already established clinical intervention to be more accessible for those in our community who are most impacted by obesity, we showed that health promotion within a healthy lifestyle programme can play a key role in seeing positive results that persist long term. The improvement in health knowledge and development of health beliefs observed in this study demonstrate a wider beneficial outcome of the programme beyond direct effects on obesity. Multidisciplinary intervention programmes may offer a real opportunity for health promotion that could support persistent healthy lifestyle change. Thus, knowledge-sharing of healthy lifestyle messages beyond the referred individual should be seen as a key strength of family-focused multidisciplinary healthy lifestyle programmes.

## Figures and Tables

**Table 1 nutrients-14-04363-t001:** Demographic characteristics of Whānau Pakari participants included in analyses of health knowledge over time (food knowledge items).

		All (Baseline)	12 Months	24 Months	60 Months
*n* ^a^		161	145 (90%)	120 (75%)	80 (50%)
Age (years)		10.5 (8.2, 13.0)	11.4 (8.9, 13.7)	12.5 (10.1, 14.7)	14.8 (12.3, 18.3)
Female		88 (55%)	75 (52%)	63 (53%)	44 (55%)
Ethnicity ^b^					
	New Zealand European	77 (48%)	70 (48%)	56 (47%)	37 (46%)
	Māori	69 (43%)	60 (41%)	51 (43%)	37 (46%)
	Asian	6 (4%)	6 (4%)	6 (5%)	2 (3%)
	European	4 (3%)	4 (3%)	4 (3%)	3 (4%)
	Pacific Islander	3 (2%)	3 (2%)	1 (1%)	nil
	Other	2 (1%)	2 (1%)	2 (2%)	1 (1%)
Deprivation quintile ^c^					
	1 (least deprived)	18 (11%)	18 (12%)	13 (11%)	7 (9%)
	2	25 (16%)	23 (16%)	22 (19%)	14 (18%)
	3	35 (22%)	33 (23%)	29 (24%)	13 (16%)
	4	38 (24%)	36 (25%)	30 (25%)	23 (39%)
	5 (most deprived)	45 (28%)	35 (24%)	26 (22%)	23 (29%)
Accompanying caregiver					
	Mother	135 (84%)	121 (83%)	102 (85%)	68 (84%)
	Father	19 (12%)	16 (11%)	12 (10%)	9 (11%)
	Other	7 (4%)	6 (4%)	6 (5%)	4 (5%)
	Not identified	nil	3 (2%)	nil	nil

Age data are medians (quartile 1, quartile 3); all other data are *n* (%). ^a^ Percentages represent the proportions of participants assessed at baseline re-assessed at a given follow-up. ^b^ Prioritised ethnicity. ^c^ Based on New Zealand Deprivation Index 2006 [18].

**Table 2 nutrients-14-04363-t002:** Number and proportion of Whānau Pakari participants who correctly answered the various items in the health knowledge questionnaire at the follow-up assessments compared to baseline.

		Baseline	12 Months	*p*	Baseline	24 Months	*p*	Baseline	60 Months	*p*
Healthy Eating										
*n*		145	145		120	120		81	81	
Healthy	Plain water	130 (90%)	140 (97%)	**0.041**	107 (90%) ^a^	119 (100%) ^a^	**<0.001**	69 (90%) ^b^	77(100%) ^b^	**0.012**
	Plain milk	105 (72%)	127 (88%)	**0.001**	88 (73%)	108 (90%)	**0.002**	59 (74%) ^c^	67 (84%) ^c^	0.13
	All fruit and vegetables	125 (86%)	137 (95%)	**0.031**	104 (87%)	114 (95%)	**0.031**	68 (85%)	77 (96%)	**0.012**
Unhealthy	Flavoured milk	88 (61%)	99 (68%)	0.17	73 (61%)	101 (84%)	**<0.001**	51 (63%)	71 (88%)	**<0.001**
	Fizzy drinks	132 (91%)	141 (97%)	**0.035**	110 (92%)	119 (99%)	**0.012**	74 (91%)	80 (99%)	0.070
	Takeaways and fast foods	120 (83%)	137 (95%)	**0.002**	101 (84%)	116 (97%)	**0.001**	68 (84%)	76 (94%)	0.057
	Crisps and chips	120 (83%)	136 (94%)	**0.001**	101 (84%)	112 (93%)	**0.035**	67 (83%)	78 (96%)	**0.003**
	Sweets and lollies	133 (92%)	142 (98%)	**0.004**	111 (93%)	118 (98%)	0.065	78 (96%)	81 (100%)	0.25
	Snack bars	55 (38%)	68 (47%)	0.12	48 (40%)	71 (59%)	**0.002**	32 (40%)	47 (58%)	**0.033**
Physical activity										
*n*		144	144		118	118		72	72	
Duration ^d^	≥60 min per day	47 (33%)	99 (69%)	**<0.001**	38 (32%)	89 (75%)	**<0.001**	22 (31%)	48 (67%)	**<0.001**

Data are number (*n*) and proportion (%) of participants with correct responses at a given assessment. Baseline data for comparison are provided for those participants with corresponding data at each follow-up assessment. *p*-values for statistically significant differences from baseline (at *p* < 0.05) are shown in bold and were derived from McNemar’s tests. ^a^ *n* = 119, ^b^ *n* = 77, and ^c^ *n* = 80 respondents. ^d^ Minimum recommend amount of physical activity per day according to the NZ Ministry of Health.

**Table 3 nutrients-14-04363-t003:** Comparison of health knowledge between Whānau Pakari participants and age-matched data from the National Survey of Children and Young People’s Physical activity and Dietary Behaviours in New Zealand 2008/09.

			Whānau Pakari					
		NZ Survey	Baseline	*p*	12 Months	*p*	24 Months	*p*
*Aged 5–9 years*								
*n*		755	98		41		29	
Healthy	Plain water	719 (96%)	85 (87%)	**0.016**	38 (93%)	0.45	29 (100%)	0.64
	Plain milk	610 (81%)	78 (80%)	0.78	37 (90%)	0.15	26 (90%)	0.33
	All fruit and vegetables	685 (91%)	87 (89%)	0.56	41 (100%)	**0.042**	26 (90%)	0.75
Unhealthy	Flavoured milk	357 (48%)	55 (56%)	0.10	28 (68%)	**0.005**	28 (97%)	**<0.001**
	Fizzy drinks	698 (93%)	89 (91%)	0.60	37 (90%)	0.55	29 (100%)	0.26
	Takeaways and fast foods	625 (83%)	78 (80%)	0.46	39 (95%)	**0.049**	27 (93%)	0.21
	Crisps and chips	602 (82%)	75 (77%)	0.48	38 (93%)	**0.043**	27 (93%)	0.10
*Aged ≥ 10 years*								
*n*		1197	134		98		89	
Duration ^a^	≥60 min per day	731 (61%)	49 (37%)	**<0.001**	73 (74%)	**0.009**	71 (80%)	**<0.001**

Data are *n* (%). *p*-values for statistically significant differences (at *p* < 0.05) are shown in bold and were derived from Chi-square tests or Fisher’s exact tests. ^a^ Minimum recommend amount of physical activity per day according to the NZ Ministry of Health; for this parameter, the comparison group were those aged 10–17 years from the NZ Survey.

**Table 4 nutrients-14-04363-t004:** Themes and subthemes identified from open-ended questions in the Whānau Pakari cohort, and respective examples as provided by study participants.

Themes	Subthemes	Benefits of Healthy Eating	Benefits of Physical Activity
Enhanced physical functioning	Overall health	“being healthy”, “stay healthy”	“health”, “healthy”, or “healthier”
	Energy levels	“gives you energy”, “having more energy”, “more sustained energy”	“higher levels of energy”, “energy levels increase”
	Bodily function	“because it won’t make your teeth rotten”, “can fight sickness”, “makes us grow”,	“strengthen calcium bones, revitalise immune system”, “good for the heart”, “helps your brain”, “good for muscles and strength”
	Fitness and strength	“stay fit”, “get fit”, “helps sports”, “faster and stronger”	“better health and fitness”, “keeps you fit”
	Life expectancy	“longer life span”, “live longer”	“healthy longer life”
	Weight loss	“don’t gain too much weight”, “lose weight”	“weight control”, “maintain healthy weight”
Enhanced mental and emotional wellbeing	Mood or perception	“you’re in a better mood”, “so you feel great”	“makes you feel good”
	Self esteem	“feeling good about yourself”, “feel better about yourself”, “[improved] confidence”.	“And you may feel good about yourself if you are active”, and “[helps you to] look good”.
	Socialising and outdoor time	n/a	“join more sports at school”, “play with friends”
Enhanced appearance	Outward appearance	“skinnier, “thinner”, “skinny tummy” (earlier timepoints) “get a better body shape”, “get slim” (later timepoints)	“better about the personal appearance and inner feelings”, “you don’t get fat”, “keep you in shape” and “skinny”.

n/a, not applicable for the given domain.

## Data Availability

The data presented in this study are available on request from the corresponding author. The data are not publicly available due to the inclusion of private health information.

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
