# Peer review of "The Impact of a Family-Based Assessment and Intervention Healthy Lifestyle Programme on Health Knowledge and Beliefs of Children with Obesity and Their Families"

_nutrients, 2022, doi:10.3390/nu14204363_

Round 1
Reviewer 1 Report
Summary:
This paper reported the health knowledge and belief of children with obesity and their families, recruited as part of a family-based assessment and intervention, at four timepoints and compared each of the follow-ups (12-, 24- and 60- months follow-up) with baseline. While the authors claimed to find a positive impact of the family-based assessment and intervention programme on health knowledge which sustained long-term, there is uncertainty about these findings due to the loss of follow-ups who may or may not have had improved health knowledge.
My specific comments are provided below:
1. Abstract: Please provide % and number (or relevant statistics) when reporting the results. I understand there are multiple outcomes, but the authors could present the key outcomes as they did in the results section. This will also help in separating the quantitative part from the qualitative part.
2. Introduction: The justification for community-based care through home-based assessment is not true. The systematic review of Sara N (https://www.ncbi.nlm.nih.gov/pmc/articles/PMC3691541/) identified 9 such studies with at least 1 study that had a community setting in combination with home.
3. Method:
1)The authors have combined trial participants from both groups as there was no observed difference in treatment outcomes. But this does not mean there will be no difference in health knowledge. Considering the high-intensity group received weekly physical or nutrition sessions in addition to home-based assessment compared to the low-intensity group, one would assume the regular follow-up would have an impact on health education. The authors should present the results in two groups (at least in supplementary if there is no difference).
2) Instead of McNemar's test or Chi-square test, the difference in proportion test would have been more appropriate as it would allow the authors to estimate an effect size (as a difference in proportion) and its variability (as a 95% confidence interval). Several articles have discussed the risk of interpreting findings based on p-values and more specifically based on p <0.05 (https://www.nature.com/articles/nmeth.4120). But I appreciate that the authors have reported the actual p-values.
3) The authors did not take the missing data into account. Discussing this as a limitation may not be enough. The author should perform a sensitivity analysis by imputing missing data and comparing the findings between imputed data and complete cases.
4) The study was conducted between 2012 to 2015. So I am assuming baseline was conducted during 2012/13, 12 months follow-up would be during 2013/14, 24 months follow-up would be during 2014/15 and 60 months follow-up would be during 2016/17 (please provide this information in the paper). The National Survey was conducted during 2008/9. So the baseline, 12- and 24- months could be compared (still 24- months follow-up would be five years after). But I am not sure about the justification for comparing data from 08/09 with that of 2016/17 (I assumed).
Results:
1) Table 1: Please provide the distribution of dyads in two groups.
2) Please report the number along with % since the denominator varies across follow-up and outcomes.
3) Table 3: Some of the baseline outcomes (i.e. plain water, crisps, and chips) are poor compared to the national survey. This needs to be discussed.
Discussion / Conclusion:
The authors should highlight the key findings (they want to discuss them) in the first paragraph.
I appreciate the discussion about the study limitation. Some of these should be discussed in the methods to prepare the reader, not to over-interpret the findings.
While there is a positive impact (according to the authors) in terms of improved health knowledge. But this did not translate into practice as the RCT did not find any difference between the two groups. Hence, the conclusion needs to provide clear guidance in terms of policy implications.
Reviewer 2 Report
- In line 86, the study states that one group only received “home-based assessment-and-advice” and it might be useful to include an example of advice in that paragraph. The assessment is described but I am a little confused on what kind of advice they are talking about.
- line 174: says 239 dyads at baseline but table 1 says baseline was 161. This is confusing.
- line 194: table 2 is a little hard to read. Suggest adding grid lines for easier visual distinction of each row/column.
- In line 248, there is a statement that says the amount of “don’t know” responses decreased. Could those frequencies be broken out in Table 2? Would be valuable to see how many changed from “don’t know” to answering correctly.
- Mentioning the Good Start programme in Queensland and the SHARE-AP ACTION in Canada in the discussion confused me a little bit. At first I thought that the Good Start program was this program that they just ran, but then I realized it is entirely different and is mentioned to bring up the inclusion of Maori and Pacific facilitators. Maybe state that these two programs were used as a model for developing interventions to be culturally sensitive?
Reviewer 3 Report
Te’o DT et al examined in the submitted manuscript the educational effectiveness of a family-based health intervention program in New Zealand on the health knowledge and beliefs of overweight or obese children and their parents. Study findings suggested participants gained knowledge on the health values of foods and beverage as well as physical activity recommendation at 12 months post intervention and such knowledge was retained at the 2-year and 5-year post evaluations. Based on the comparison between study results and national survey data, this intervention program significantly elevated participants health knowledge at 12-month follow-up assessment.
The research questions addressed in this paper align well with the scope of the journal. The statistical analyses were appropriate to evaluate the retention of knowledge over time and the comparisons with national survey data. Results are presented clearly in tables and provide adequate evidence for study conclusions.
Specific comments on Table 1 (demographic characteristics):
1. 1. The authors stated that the questionnaires were completed by child-caregiver dyads at every follow-up assessment. What proportion of the caregivers was mothers (vs. fathers completing the questionnaires) very time? This information will add insight to why this interventional program was successful.
2. 2. At every follow-up visit, what was the average educational attainment of the caregivers who responded to the questionnaires? The authors mentioned in the Discussion section that participating caregivers might have been more engaged and motivated. Knowing their education level will help shed light on such speculation.
Author Response
Overall comments: Te’o DT et al examined in the submitted manuscript the educational effectiveness of a family-based health intervention program in New Zealand on the health knowledge and beliefs of overweight or obese children and their parents. Study findings suggested participants gained knowledge on the health values of foods and beverage as well as physical activity recommendation at 12 months post intervention and such knowledge was retained at the 2-year and 5-year post evaluations. Based on the comparison between study results and national survey data, this intervention program significantly elevated participants health knowledge at 12-month follow-up assessment.
The research questions addressed in this paper align well with the scope of the journal. The statistical analyses were appropriate to evaluate the retention of knowledge over time and the comparisons with national survey data. Results are presented clearly in tables and provide adequate evidence for study conclusions.
Reply: Thank you very much for your time taken to appraise our manuscript and provide valuable feedback.
Questions/comments:
- Specific comments on Table 1 (demographic characteristics):
The authors stated that the questionnaires were completed by child-caregiver dyads at every follow-up assessment. What proportion of the caregivers was mothers (vs. fathers completing the questionnaires) very time? This information will add insight to why this interventional program was successful.
Reply: The vast majority of caregivers were mothers, who made up a consistent proportion of accompanying caregivers throughout the study (83–85%). In light of the reviewers, comment we have since added these data into all tables providing demographic characteristics for our study population. We have also added a statement to this regard at the end of the 1st paragraph of the Results section:
“At all time points, the vast majority of accompanying caregivers were mothers (83–85%), with fathers making up an additional 10–12% (Table 1).”
- Specific comments on Table 1 (demographic characteristics):
At every follow-up visit, what was the average educational attainment of the caregivers who responded to the questionnaires? The authors mentioned in the Discussion section that participating caregivers might have been more engaged and motivated. Knowing their education level will help shed light on such speculation.
Reply: Unfortunately, limited information was collected on the caregivers – we originally had questions relating to household income and educational attainment, however the majority of participants did not want to answer these questions. Since the Whānau Pakari trial was conducted within a clinical service focussed on the health of the children, we did not want to offend participants with continuing to ask these questions and potentially dissuade them from the service. Therefore, we made provision of caregiver information completely optional, and while we were able to collect information on the caregiver's relationship to the child, we did not gather comprehensive data on educational attainment of the caregiver.
In light of the Reviewer's comment, we have since included this issue among the limitations of our study (Discussion, last paragraph):
"(…) The nature of this study as an analysis of a real-world health service also limited the amount of data available for caregivers, for whom providing information about themselves during the assessment was not a requirement for participation. As a result, potentially important data such as caregiver's educational attainment were not collected."